# The Roles of Proton-Sensing G-Protein-Coupled Receptors in Inflammation and Cancer

**DOI:** 10.3390/genes15091151

**Published:** 2024-09-01

**Authors:** Calvin R. Justus, Mona A. Marie, Edward J. Sanderlin, Li V. Yang

**Affiliations:** Department of Internal Medicine, Brody School of Medicine, East Carolina University, Greenville, NC 27834, USA

**Keywords:** inflammation, cancer, proton-sensing GPCRs, GPR4, GPR65, GPR68

## Abstract

The precise regulation of pH homeostasis is crucial for normal physiology. However, in tissue microenvironments, it can be impacted by pathological conditions such as inflammation and cancer. Due to the overproduction and accumulation of acids (protons), the extracellular pH is characteristically more acidic in inflamed tissues and tumors in comparison to normal tissues. A family of proton-sensing G-protein-coupled receptors (GPCRs) has been identified as molecular sensors for cells responding to acidic tissue microenvironments. Herein, we review the current research progress pertaining to these proton-sensing GPCRs, including GPR4, GPR65 (TDAG8), and GPR68 (OGR1), in inflammation and cancer. Growing evidence suggests that GPR4 and GPR68 are mainly pro-inflammatory, whereas GPR65 is primarily anti-inflammatory, in various inflammatory disorders. Both anti- and pro-tumorigenic effects have been reported for this family of receptors. Moreover, antagonists and agonists targeting proton-sensing GPCRs have been developed and evaluated in preclinical models. Further research is warranted to better understand the roles of these proton-sensing GPCRs in pathophysiology and is required in order to exploit them as potential therapeutic targets for disease treatment.

## 1. Introduction

Localized tissue acidosis is a main characteristic of inflammatory milieus. Numerous studies have demonstrated that a local tissue pH below 7.0, and sometimes even below 6.0, is detected in inflamed tissues, which can impact cellular function [1,2,3,4]. The acidic microenvironment in inflamed tissue occurs predominately due to the increased metabolic demand from infiltrating immune cells and hypoxia. These immune cells increase the oxygen consumption and glucose uptake for glycolysis and oxidative phosphorylation. When oxygen availability is limited, cells often undergo anaerobic glycolysis. This process generates increasing amounts of lactic acid, thereby creating a local acidic microenvironment within inflammatory loci [1]. However, other factors can contribute to these acidic inflammatory loci. Neutrophils are often the first responders to the site of inflammation for the elimination of pathogens such as bacteria. Bacteria can acidify the inflammatory microenvironment due to the accumulation of short-chain fatty acids from microbial metabolism [5,6]. Neutrophils and macrophages can attempt to eliminate pathogenic bacteria through respiratory bursts, which can further deplete oxygen and acidify the microenvironment. Indeed, dysregulated local pH is a hallmark of inflamed tissue microenvironments [1,2,3,4].

Acidosis is not merely limited to inflammatory states in tissue but is also a hallmark of the tumor microenvironment [7,8,9]. Tumor extracellular pH values can range between pH 6.4 and pH 7.0, which is markedly acidic when compared to the tightly regulated physiological pH values of 7.3 to 7.4 found in normal tissue. This phenomenon can be partly explained by poor tumor blood perfusion, as blood perfusion is required for the delivery of oxygen and the removal of metabolic byproducts in the context of highly proliferative tumor cells. As such, neovascularization within the tumor microenvironment can be hampered and result in elevated vascular permeability due to vessel structural deficiencies. This can produce spatially hypoxic regions in the tumor and influence cellular metabolic programs unique to highly metabolically active cancer cells. It is well documented that tumor cells utilize both glycolysis and oxidative phosphorylation for energy production, but preferentially shift metabolic activity towards glycolysis, even in the presence of oxygen (the Warburg effect) [10,11,12,13]. Terminal byproducts of elevated tumor-cell glycolytic metabolism, such as protons and lactate, are efficiently pumped into the extracellular space, reducing the tumor pH.

Acidosis has profound biological effects and can regulate various cellular and physiological processes, such as cell proliferation, differentiation, and death; the inflammatory response; immune cell function; angiogenesis; tumor formation; therapeutic responses; nociception; and others [1,7,14,15,16,17,18]. Extracellular acidosis can be sensed by cells through several types of ion channels and receptors [9,19,20].

## 2. Overview of Proton-Sensing GPCRs

Extracellular acidosis is a common pathophysiological stressor that develops in the microenvironment of inflamed tissues and tumors. There are several molecular mechanisms activated by extracellular acidosis that support cellular adaptation to the changing microenvironment. Cells can detect an acidic extracellular pH through acid-sensing ion channels (ASICs) and transient receptor potential (TRP) channels [19]. Extracellular acidosis also stimulates inositol polyphosphate formation and promotes calcium mobilization in various cell types [21,22,23,24,25,26]. In contrast, intracellular acidosis does not promote calcium mobilization, indicating the presence of a cell-surface receptor responsible for modulating cellular activity in response to extracellular acidosis. It was hypothesized that acid-sensing receptors are activated by a certain functional group, specifically the imidazole of a histidine residue [21,22]. These acid-activated receptors were later revealed to be a family of proton-sensing GPCRs, including GPR4, GPR65, and GPR68 [27]. As hypothesized in [21], studies established that proton-sensing GPCRs sense extracellular pH via the protonation of histidine residues on the extracellular domain of the receptor [27,28]. These proton-sensing GPCRs can be partially activated within the physiological pH range of 7.3–7.4, but peak activation occurs between pH 6.4 and pH 7.0. The activation of proton-sensing GPCRs initiates downstream signaling through the G_q/11_, G_s_, and G_12/13_ pathways. GPR65, GPR4, and GPR68 couple to G_s_ and G_12/13_, while GPR4 and GPR68 can also couple to G_q/11_ [29]. Overall, proton-sensing GPCRs are a novel family of receptors that detect extracellular acidosis and modulate cellular function in response to it.

The family of proton-sensing GPCRs contains distinct expression profiles. GPR4 is highly expressed in vascular-rich tissues such as the lung, liver, and kidney, as well as soft tissues [30,31,32]. In line with these observations, GPR4 is predominately expressed in vascular endothelial cells [33,34]. Recent studies have shown that GPR4 is also expressed in neurons of the retro-trapezoid nucleus (RTN) and white adipose tissue [29]. GPR65 expression is highest in leukocyte-rich tissues like the spleen, bone marrow, and lymph nodes, owing to its predominate expression in immune cells [35,36,37,38]. GPR65 is expressed in both myeloid- and lymphoid-derived cells. Recent studies have also demonstrated a role for GPR65 expression in neurons [16,29]. GPR68 is expressed broadly in various tissues, such as the spleen, brain, lung, heart, and kidney [39]. GPR68 has also been investigated in the immune system as GPR68 is expressed in dendritic cells, macrophages, neutrophils, and T cells. However, GPR68 is also expressed on and has regulatory functions in fibroblasts, dorsal root ganglia, osteoclasts, and cardiomyocytes [29].

Given the diverse expression of the proton-sensing GPCR family and its distinct G-protein activation status, each member has been implicated in a variety of physiological systems [20,29,40]. Studies have suggested roles for GPR65 in the respiratory system (asthmatic inflammation), nervous system (nociception and panic disorders), skeletal system (bone resorption/density), and immune system (leukocyte inflammatory responses). GPR68 has been studied in the cardiovascular (cardiomyocyte viability), renal (acid–base homeostasis), respiratory (inflammatory airway remodeling), gastrointestinal (intestinal homeostasis), skeletal (bone acid sensing), and endocrine (insulin secretion) systems. Furthermore, GPR4 has been investigated in the nervous system (CO_2_ chemosensing), endocrine system (insulin sensitivity), renal system (acid–base balancing), cardiovascular system (angiogenesis), and immune system (inflammation) [20,29,40]. In addition to their physiological functions, proton-sensing GPCRs also play important roles in a wide range of diseases, such as cancer and inflammatory disorders [20,29,40].

## 3. Roles of GPR4 in Inflammation and Cancer

### 3.1. GPR4 in Inflammatory Disorders

GPR4 regulates the inflammatory response primarily through the mediation of vascular endothelial cell activation and endothelium–leukocyte interaction. Endothelial cells compose the inner layer of blood vessels that often penetrate acidic tissue microenvironments, especially in the inflammatory loci and tumors. Among the proton-sensing GPCR family, GPR4 has the highest expression in endothelial cells. Several studies have demonstrated that GPR4 can regulate endothelial cells’ inflammatory responses in acidic tissue microenvironments [14,33].

The pro-inflammatory role of GPR4 has been established in several organ systems, such as the nervous, cardiovascular, respiratory, renal, skeletal, skin, and gastrointestinal systems, and in cancer [29,34,41,42,43,44,45,46,47,48,49]. Upon acidic activation, GPR4 mediates its function through downstream G_s_/cAMP, G_12/13_/Rho, and G_q/11_/PLC pathways [28,50]. In vitro acidic activation of GPR4 leads to endothelial cell activation, regulating their adhesiveness and leukocyte adhesion through G-protein pathways, including the G_s_/cAMP/Epac pathway [14]. This activation further augments the endothelium pro-inflammatory gene expression of *IL-8*, *CCL20*, *TNF-α*, *COX-2*, and *IL-1A*, adhesion molecules *VCAM-1* and *E-selectin* [33], and ER stress response genes, *PERK*, *ATF6*, *ATF3*, *CHOP*, and *IRE1* [45]. Moreover, GPR4 also functions through G_12/13_/Rho GTPase pathways, leading to vascular leakiness due to induced actin stress fiber formation [41]. Under acidic conditions, GPR4 has been found to be implicated in increasing the expression of RANK/RANKL/OPG in osteoblastic cells [51]. Additionally, GPR4 overexpression promotes the development of osteoarthritis (OA), and the pharmacological inhibition of GPR4 protects mice against OA via regulating the CXCL12/CXCR7 pathway [44]. GPR4 expression was found to be upregulated in IL-33-activated human mast cells, and pharmacological inhibition of GPR4 countered the pro-inflammatory action of IL-33 by downregulating *IL-17*, *IFN-γ*, and *TNF-α* [52]. GPR4 has also been associated with increased nociception of the dorsal root ganglia [16], and cAMP-mediated upregulation of the RANK/RANKL/OPG system and neurotrophins by nucleus pulposus cells, thus promoting intervertebral disk degeneration [53]. GPR4 expression was found to be upregulated in cardiomyocytes by oxygen–glucose deprivation/reoxygenation (OGD/R) and GPR4 knockdown reduced the NLRP3 inflammasome signaling and attenuated OGD/R-induced cardiomyocyte inflammation and apoptosis [54]. Transcriptome-wide association studies (TWASs) have identified GPR4 as a biomarker in Alzheimer’s patients [55,56]. Pharmacological inhibition and genetic knockout of GPR4 protects against neuronal cell death through the modulation of PIP2 degradation-mediated calcium signaling, thereby protecting against neurodegenerative disorders such as Parkinson’s disease [42]. Moreover, mice lacking GPR4 showed enhanced glucose tolerance and insulin sensitivity [57], and lower blood pressure [58]. Most recently, a pro-inflammatory role for GPR4 has been proposed in COVID-19 [48], where GPR4 gene expression was found to be upregulated in COVID-19 patients’ lung and colon samples by 2.3-fold and 3.9-fold, respectively [47,48]. Additionally, a correlation has been identified between GPR4 RNA expression obtained from patient nasal swabs at the time of testing and COVID-19 severity at a later stage [49].

It has been observed that GPR4 expression is upregulated in the colon of ulcerative colitis (UC) and Crohn’s disease (CD) patients [34,59]. In addition, GPR4 is positively correlated to increased fibrosis in CD patients [60]. We and others have shown the pro-inflammatory role of GPR4 in acute and chronic inflammatory bowel disease (IBD) mouse models [34,46,59]. GPR4 confers a pro-inflammatory function in IBD through modulating chemokines, cytokines, and adhesion molecule expression on endothelial cells, regulating leukocyte infiltration into the intestinal tissues of IBD mouse models [34]. Pharmacological inhibition of GPR4 mitigates IBD in mouse models and downregulates cytokines such as *TNF-α*; adhesion molecules such as *VCAM-1*, *E-selectin*, and *MAdCAM-1*; and fibrosis markers such as *COL1A1/2* [46,60]. Overall, pharmacologically antagonizing GPR4 confers anti-nociceptive, anti-fibrotic, anti-angiogenic, and anti-inflammatory effects [46,60,61,62].

### 3.2. GPR4 in Tumor Biology

GPR4 has been investigated in various types of cancer such as melanoma, epithelial ovarian carcinoma, head-and-neck cancer, and colon cancer [63,64,65,66,67,68,69]. Depending on the cancer type and context, both pro- and anti-tumorigenic effects of GPR4 have been reported. One study found that overexpression of GPR4 in NIH3T3 fibroblasts stimulated cell transformation [65]. Overexpression of GPR4 in SCCHN (squamous-cell carcinoma of the head and neck) cells increased the expression and secretion of pro-angiogenic factors IL6, IL8, and VEGFA and promoted angiogenesis in vitro [67]. GPR4 expression was found to be upregulated in colorectal tumor samples and GPR4 signaling promoted colorectal cancer cell proliferation and metastasis through the Hippo pathway [69]. Alternatively, overexpression of GPR4 in mouse B16F10 melanoma cells reduced cell migration and metastasis, although primary tumor growth was not significantly affected [63,70]. In melanocytes, upon overexpression of the BRAF^V600E^ oncogene, disruption of GPR4 was found to be highly selected, suggesting a tumor-suppressor role [68]. Overall, GPR4 has been reported to diminish and promote malignancy, which is dependent on cancer and cell type.

In addition to cancer cells, the function of GPR4 in stromal cells in the tumor microenvironment has been studied. GPR4 is highly expressed in vascular endothelial cells and plays a role in tumor angiogenesis. Some researchers found that GPR4 expression was positively correlated with microvascular density in ovarian carcinoma [66]. The angiogenic response to VEGF, allograft tumor growth, and angiogenesis were all found to be reduced in GPR4-deficient mice [64]. Also, GPR4 deficiency curtailed tumor growth and angiogenesis in one colitis-associated colorectal cancer mouse model [71]. A recent study showed that activation of GPR4 by an acidic pH increased the expression of inflammatory chemokines and adhesion molecules in lymphatic endothelial cells and promoted cancer cell metastasis, which were both attenuated by GPR4 knockdown and inhibition [72]. Altogether, the current literature suggests that the inhibition of GPR4 has anti-inflammatory and anti-angiogenic effects.

### 3.3. GPR4 Small-Molecule Modulators

GPR4 small-molecule inhibitors have been developed and their biological effects have recently been characterized (Table 1). Derivatives of imidazo-pyridine and pyrazolopyrimidine compounds have been identified as novel GPR4 inhibitors, such as GPR4 antagonist 13 (NE-52-QQ57), 2-(4-((2-Ethyl-5,7-dimethylpyrazolo [1,5-a]pyrimidin-3-yl)methyl)phenyl)-5-(piperidin-4-yl)-1,3,4-oxadiazole, and related compound 39C [61,62,73,74]. The GPR4 antagonist 39C (EIDIP), 2-Ethyl-3-{4-[(E)-3-(4-isopropyl-piperazin-1-yl)-propenyl]-benzyl}-5,7-dimethyl-3H-imidazo [4,5-b]pyridine, was found to reduce the expression of inflammatory chemokines, cytokines, adhesion molecules, NF-κB pathway genes, and stress-responsive genes, which include *IL-1*, *IL-8*, *CXCL1*, *CXCL2*, *CCL2*, *CCL7*, *VCAM-1*, *ICAM-1*, *E-selectin*, *RELB*, *COX2*, *ATF3*, and *CHOP* in primary-cultured endothelial cells [33,45]. Several studies have shown that GPR4 antagonists reduce inflammation in antigen-induced arthritis rat models, dextran sulfate sodium (DSS)-induced acute colitis mouse models, and short-term emphysema exacerbation COPD mouse models, and can alleviate inflammatory pain in a complete Freund’s adjuvant-induced hyperalgesia rat model [46,61,62,75]. In the DSS-induced acute colitis model, NE-52-QQ57 ameliorated intestinal inflammation and decreased the expression of TNF-α in the inflamed mouse colon tissues [46]. In an IL10-deficient mouse spontaneous colitis model, NE-52-QQ57 did not exhibit significant efficacy but demonstrated a trend towards improvement for several disease parameters such as histopathology, inflammation, and hyperplasia scores [76]. Similarly to GPR4 genetic knockout mice, NE-52-QQ57 exhibited anti-angiogenic effects and attenuated inflammatory responses, tissue edema, and exudate formation [41,61,62]. Furthermore, treatment with NE-52-QQ57 inhibited the expression of inflammatory molecules including TNF-α, IL-1β, IL-6, iNOS, nitric oxide (NO), COX2, and PGE2 in cultured chondrocytes [77]. The GPR4 antagonist NE-52-QQ57 alleviated mouse osteoarthritis progression and promoted extracellular matrix production [44]. Another report found that treatment with the GPR4 antagonist 39C reduced fibrogenesis in a heterotopic intestinal transplant model to the levels observed with pirfenidone [60]. Furthermore, studies have demonstrated that genetic knockout and pharmacological inhibition of GPR4 protect mice from ischemic injury in myocardial infarction, renal ischemia–reperfusion, and hindlimb ischemia–reperfusion mouse models [41,73,78]. Recently, a study demonstrated that the GPR4 antagonist NE-52-QQ57 reduced brain hemorrhagic lesions, cerebral edema, and arteriovenous malformations in a GPRASP1 loss-of-function mouse model [79]. It has also been shown that NE-52-QQ57 attenuated dopaminergic neuronal loss and ameliorated motor and memory functions in the 1-methyl-4-phenyl-1,2,3,6-tetrahydropyridine (MPTP)-treated mouse model of Parkinson’s disease [80]. In the area of tumor biology, it was shown that a GPR4 antagonist (3b) partially suppressed acidic-pH-induced expression of the growth hormone and prolactin in MtT/S pituitary tumor cells [81]. Treatment with the GPR4 antagonist NE-52-QQ57 reduced the lymphangiogenesis and lymph-node metastasis of melanoma cells in a mouse cancer metastasis model [72]. Overall, it has been shown that GPR4 small-molecule inhibitors exhibit anti-inflammatory, anti-fibrotic, anti-nociceptive, anti-angiogenic, and anti-tumorigenic effects in various in vitro and in vivo models.

## 4. Roles of GPR65 in Inflammation and Cancer

### 4.1. GPR65 in Inflammatory Disorders

GPR65 was originally identified by cloning as an orphan GPCR that was observed to be upregulated during thymocyte apoptosis; therefore, this receptor is also known as T-cell-death-associated gene 8 (TDAG8) [35,82]. Large-scale genome-wide association studies (GWASs) have identified GPR65 as a susceptibility gene associated with several inflammatory diseases such as IBD, multiple sclerosis, asthma, atopic dermatitis, heparin-induced thrombocytopenia, and spondyloarthritis [83,84,85,86,87,88,89,90]. GPR65 is highly expressed in leukocytes and regulates the inflammatory response of immune cells [91,92,93,94]. It was demonstrated that GPR65 inhibited pro-inflammatory cytokine secretion, which includes IL-6 and TNF-α, in mouse peritoneal macrophages upon activation by extracellular acidification. This cytokine inhibition was shown to occur through the G_s_/cAMP/protein kinase A (PKA) signaling pathway [93,95]. Treatment with dexamethasone, a potent glucocorticoid, increased GPR65 expression in peritoneal macrophages. Following dexamethasone treatment, there was an inhibition of TNF-α secretion in a manner dependent on an increased expression of GPR65 [91]. Furthermore, a GPR65 agonist, BTB09089 (3-[(2,4-dichlorobenzyl)thio]-1,6-dimethyl-5,6-dihydro-1H-pyridazino[4,5-e][1,3,4]thiadiazin-5-one), was able to inhibit lipopolysaccharide (LPS)-induced macrophage pro-inflammatory cytokine expression and anti-CD3 splenocyte stimulation [94]. Another study investigated the role of GPR65 and acidosis in microglia inflammatory responses [92]. GPR65 was shown to reduce IL-1β secretion through the G_s_/cAMP/PKA pathway. Additional studies showed that GPR65 alleviated myocardial infarction-induced inflammation through inhibiting resident macrophage secretion of CCL20, a chemokine for γδT cells [96]. Another study found that genetic deletion of GPR65 exacerbated inflammation in a type II collagen-induced mouse arthritis model [94,97]. Furthermore, GPR65 deletion was investigated in an experimental autoimmune encephalomyelitis mouse model. The data showed that GPR65 reduced disease severity presumably through invariant natural killer T cells [98]. In an LPS-induced acute lung injury mouse model, GPR65 attenuated neutrophil accumulation and lung damage by inhibiting chemokine production [99].

In an acidic microenvironment, the expression of GPR65 by immune cells regulates the inflammatory response by a myriad of differential mechanisms, such as its role in the Th1/Th17 differentiation of CD4+ T cells [100] and its role in decreasing leukocyte infiltration to the site of inflammation in mouse models of IBD and colitis-associated colorectal cancer [36]. In patients with Immunoglobulin A nephropathy (IgAN), a type of glomerulonephritis, GPR65 expression in macrophages was negatively correlated to disease severity [101]. Under persistent acidic conditions, microglia cells show impaired phagocytic function associated with a decrease in cyclic AMP (cAMP) and PKA associated with a decrease in GPR65 expression [102]. The expression of GPR65 in cardiovascular diseases confers a protective action against myocardial infarction through transcriptional downregulation of the chemokine CCL20’s expression in macrophages, which, in turn, downregulates the production of pro-inflammatory IL17A [96]. Similarly, GPR65 can protect against brain ischemia–reperfusion injury by inhibiting microglia inflammatory activation in a transient middle cerebral artery occlusion (tMCAO) mouse model [103].

It has been reported that the rs8005161 polymorphism of GPR65 may alter the response of human monocytes towards acidic pH activation, leading to increased disease severity in human patients with IBD [88]. GPR65 polymorphism with a substitution of isoleucine to leucine at codon 231 was identified as a risk factor in colitis. Ile231Leu-GPR65 knock-in mice were challenged with bacteria and T-cell-mediated colitis and showed increased susceptibility to colitis [104]. Decreased expression of GPR65 led to impaired bacterial phagocytosis and increased pro-inflammatory signaling via the NLRP3 inflammasome in macrophages, through the G_12/13_ pathway [105]. The lack of GPR65 expression in macrophages resulted in an increased mRNA expression of pro-inflammatory cytokines such as IFN-γ, TNF, IL-6, and iNOS in murine colonic tissues, leading to a heightened course of IBD [88]. GPR65 deficiency aggravated intestinal inflammation and promoted colitis-associated colorectal cancer development in IBD mouse models [36,38,88,106]. Moreover, a recent study showed that reduced GPR65 expression in intestinal epithelial cells isolated from IBD patients correlated with reduced antimicrobial mechanisms, e.g., the production of antimicrobial peptides Reg3γ and Reg3β [107]. In addition, a reduced expression of GPR65 in intestinal epithelial cells in mice aggravated DSS- and *C. rodentium*-induced colitis through the disruption of epithelial antimicrobial responses [107]. Overall, these studies demonstrate through various mechanisms that there is an inverse correlation between GPR65 expression and colitis severity.

Other reports provide a pro-inflammatory role for GPR65 during inflammation. GPR65 was reported to increase eosinophil viability in the acidic microenvironment by reducing apoptosis through the cAMP pathway [108]. As eosinophils are central in asthmatic inflammation and allergic airway disease, GPR65 may play a role in increasing asthmatic inflammation. Notably, GPR65 appears to be involved in regulating Th17 pathogenicity. One study demonstrated that the absence of GPR65 reduced the differentiation of IL-17A+ cells in vitro [109]. Furthermore, this group reconstituted wild-type (WT) and GPR65-knockout (KO) CD4+ T cells into RAG-deficient mice and observed the loss of GPR65 in CD4+ T cells protected mice from experimental autoimmune encephalomyelitis (EAE) [109]. Another study analyzing the transcriptome signatures of Th17 cells involved in spondylarthritis suggested that GPR65 expression in Th17 cells likely contributes to enhancing GM-CSF expression and subsequent spondylarthritis disease severity [85]. In a rheumatoid arthritis (RA) mouse model, GPR65 deficiency reduced hyperalgesia and RA score and significantly decreased IL-6 and IL-17 production. These observations were associated with reduced M1 macrophages at the site of the inflamed joint and satellite glial cells in the dorsal root ganglia in the GPR65-null mice [110]. Moreover, conditional knockout of GPR65 in CD4+ T cells reduced Th1 and Th17 cell immune responses in colon mucosa and alleviated intestinal inflammation in murine colitis models [100].

### 4.2. GPR65 in Tumor Biology

GPR65 can also impact cancer development. GPR65 expression promotes cancer development in several solid cancers. For instance, GPR65 overexpression in Lewis lung carcinoma (LLC) cells increased primary tumor growth in a murine model and promoted resistance to acidosis-mediated cell death through PKA/ERK signaling [111]. GPR65 knockdown in human non-small-cell lung cancer reduced cell survival in response to acidosis [111]. In addition, GPR65 gene expression can transform immortalized mammary epithelial cells [65]. According to these reports, GPR65 is oncogenic in epithelial tumors. In contrast, GPR65 inhibits cancer progression in osteosarcoma. A recent single-cell RNA-Seq study indicated that the expression of GPR65 is reduced significantly in osteosarcoma. In addition, this lower expression of GPR65 was found to be correlated with an increased risk of metastasis and poor prognosis for osteosarcoma patients, likely by suppressing immune escape and osteosarcoma cell proliferation [112].

The role of GPR65 in blood cancer progression is less clear. GPR65 was initially discovered as a gene highly expressed during glucocorticoid-induced apoptosis of immature thymocytes and has been mapped to chromosome 14q31-q32.1, a location at which abnormalities associated with T cell lymphoma and leukemia are found [35,82]. The expression of GPR65 promotes the evasion of acidosis-induced apoptosis in WEHI7.2 and CEM-C7 T cell lymphoma cells under glutamine starvation conditions [113]. GPR65 gene expression also correlates with pro-survival members of the B cell lymphoma-2 (Bcl-2) family of mitochondrial proteins in chronic lymphocytic leukemia (CLL) [114]. Alternatively, GPR65 gene expression promotes glucocorticoid-induced apoptosis in murine lymphoma cells as well as reducing c-myc oncogene expression in human lymphoma cells [115,116]. In addition, the expression of GPR65 mRNA is reduced in follicular lymphoma and diffuse large B cell lymphoma in comparison to normal immune tissue by 50% [115]. These few reports demonstrate that GPR65 can promote and inhibit blood cancer progression, which is dependent on cell type and context. However, GPR65 function has not been adequately investigated in blood cancers. One comprehensive analysis revealed that GPR65 gene expression is reduced in the majority of blood cancers in comparison to normal immune cells or tissue [117]. To test the hypothesis that GPR65 is a tumor suppressor in blood cancers, its expression was restored in U937 acute myeloid leukemia cells, among others. Restoring GPR65 gene expression in U937 cells inhibited cell proliferation in vitro and reduced tumor growth and metastasis in vivo [117]. The outcome that precipitated from restoring GPR65 gene expression clearly confirmed that it inhibits blood cancer progression. These results are even more intriguing when considering that this study also provides evidence demonstrating that extracellular acidosis strengthened the effects observed. Overall, this study confirms the notion that GPR65 gene expression provides a selective pressure against blood cancer cells [117].

GPR65 can also modulate the tumor immune microenvironment (TIME). It was shown that GPR65 sensed tumor acidosis to induce the polarization of tumor-associated macrophages (TAMs) towards a non-inflammatory phenotype and immunoevasion [118]. Acidity in the tumor microenvironment upregulated the expression of the immune checkpoint molecule PD-L1 through the proton-sensing GPCRs, including GPR65 [119]. A recent study also showed that GPR65 on TAMs sensed lactate in the tumor microenvironment in order to induce the secretion of HMGB1 through the cAMP/PKA/CREB pathway to exacerbate glioma progression [120]. It has been proposed that GPR65 functions as a potential immune checkpoint in the tumor microenvironment [118,121].

### 4.3. GPR65 Small-Molecule Modulators

There are several agonists and antagonists that have been developed to support the modulation of GPR65 activity, and the biological effects associated with these small molecules have been investigated (Table 2). BTB09089, also known as 3-[(2,4-dichlorobenzyl)thio]-1,6-dimethyl-5,6-dihydro-1H-pyridazino[4,5-e][1,3,4]thiadiazin-5-one, is a GPR65 specific agonist that has been found to activate GPR65 to induce cAMP production [94]. BTB09089 binds in an allosteric manner to GPR65 [122]. In wild-type (WT) peritoneal macrophages, when stimulated with thioglycolate, BTB09089 suppressed the production of TNF-α and IL-6. However, cells from GPR65-knockout (KO) mice did not exhibit the same response to BTB09089, indicating that the small molecule acts through GPR65 to decrease the production of TNF-α and IL-6 [94]. BTB09089 has been used in various studies demonstrating anti-inflammatory properties. One group demonstrated that pre-treating rats with BTB09089 prior to ischemia and stroke had a protective effect against neuronal apoptosis and neurological deficits by activating the Akt pathway [123]. Additionally, delayed BTB09089 treatment inhibited microglial activation, reduced neuronal damage, and improved neurological function recovery in an ischemic stroke mouse model [124]. In contrast, BTB09089 also promoted M1 macrophage polarization through the activation of GPR65 and was able to stimulate the release of TNF-α, IL-6, and transforming growth factor-β (TGF-β) in hepatic macrophages, resulting in increased hepatic fibrosis in a murine model [125,126]. ZINC62678696 can also bind to GPR65, but inhibits the activity of the receptor [122]. Treatment of M1 macrophages with ZINC62678696 abrogated M1 macrophage polarization and reduced the release of TNF-α, IL-6, and TGF-β in hepatic macrophages [125,126]. A separate study developed GPR65-positive allosteric modulators (PAMs; e.g., BRD5075 and BRD5080) and demonstrated that activation of GPR65 by the PAMs reduced the expression of inflammatory cytokines and chemokines in dendritic cells [127]. Recently, a small-molecule inhibitor of GPR65, PTT-4256, has been developed [128]. It was shown that PTT-4256 counteracted acidic-pH-mediated immunosuppressive transcriptional programs in immune cells and had significant therapeutic efficacy in MC38 colon cancer and B16F10 melanoma mouse models [128]. Due to the biological relevance of GPR65 in inflammation and human diseases, additional studies supporting the modulation of this receptor are warranted.

## 5. Roles of GPR68 in Inflammation and Cancer

### 5.1. GPR68 in Inflammatory Disorders

GPR68 is similar to GPR4 in that its acidosis-induced activation results in enhanced inflammation; however, whether GPR68 plays a pro-inflammatory or protective role against inflammatory-induced damage is context-dependent. GPR68 is expressed in immune cells [129], fibroblasts [130], smooth-muscle cells [131], vascular endothelial cells [129,132,133], cortical neurons [134], hypothalamic neurons [135], cartilage (chondrocytes) [136], bone cells (osteoblasts and osteoclasts) [137,138,139], and epithelial cells in the proximal tubule of the kidney [133]. Studies have shown that GPR68 contributes to macrophage and dendritic cell inflammation [29,95,140,141]. Notably, GPR68 has been implicated in increasing intestinal inflammation through macrophage inflammatory programs [141,142,143]. Genetic deletion of GPR68 has been found to lead to reduced disease severity concerning intestinal inflammation in a colitis mouse model [106,141]. In another study, an increase in GPR68 mRNA expression in the inflamed colon of IBD patients was positively correlated to disease severity and clinical score [129]. Moreover, pharmacological inhibition of GPR68 using the small-molecule antagonist GPR68-I showed a mild decrease in disease severity, histopathologic parameters, macrophage infiltration, and macrophage proliferation in an acute intestinal inflammation DSS mouse model. In a chronic DSS mouse model, pharmacological inhibition showed a more significant decrease in both macroscopic and microscopic disease indicators, leading to levels comparable to those observed in GPR68-KO mice. This inhibitor also significantly decreased the expression of the tissue inflammatory indicators IL-6 and TGF-β in this chronic DSS model [144]. The same group also showed that GPR68-I inhibited GPR68-mediated inositol phosphate (IP) production in fibroblasts and CD14+ human monocytes following treatment with extracellular acidosis [129]. In WT primary murine fibroblasts, acidosis-mediated GPR68 activation resulted in IP production and RhoA-mediated F-actin stress fiber formation, which was significantly diminished in the GPR68-KO cells [129]. GPR68 expression was positively correlated to pro-fibrotic gene expression in the terminal ileum of human tissues excised from IBD patients, and its deletion protected against DSS-induced fibrosis in GPR68-KO mice [142].

GPR68 protein levels were reduced in idiopathic pulmonary fibrosis (IPF) patient lung tissues compared to healthy controls [130]. It was shown that GPR68 expression inhibited TGF-β-induced myofibroblast differentiation [130]. Consistently, GPR68 activation induced by the positive allosteric modulator Ogerin inhibited and partially reversed TGF-β-mediated myofibroblast differentiation through G_s_ activation in the primary human lung fibroblasts of healthy and IPF patients [145]. Real-time PCR also showed GPR68 expression in airway smooth-muscle (ASM) cells [131]. GPR68 siRNA knockdown significantly abolished acid-induced signaling and ASM contractility [131]. Interestingly, genetic deletion of GPR68 in a dust-mite-induced airway hyperresponsiveness (AHR) mouse model did not protect against allergen-induced lung inflammation. However, its G_s_-biased agonist, Sulazepam, but not the balanced agonist Lorazepam, showed a prophylactic effect in the wild-type AHR mice by significantly reducing airway resistance [146]. Thus, GPR68 may confer differential roles through biased signaling, and the exact role of GPR68 in the context of lung disease remains to be elucidated.

Metabolic acidosis has a role in bone resorption and can result in bone calcium efflux, leading to its excretion into the urine [147]. GPR68 is expressed in early osteoclastogenesis and is required for extracellular acid sensation in the osteoblasts [137,139]. In GPR68-KO mice, decreased bone resorption and increased bone density, with increases in the numbers of both osteoblasts and osteoclasts, have been observed [138]. Osteoblast acid sensing by GPR68 is mediated through the GPR68/G_q/11_/phospholipase C/protein kinase C pathway [137]. Moreover, GPR68 expression was significantly upregulated in the cartilages isolated from osteoarthritis (OA) patients and in a surgical OA mouse model compared to healthy individuals. This increase was positively correlated to higher-grade OA and the expression of members of the matrix metalloprotease family—namely the *MMP13*, *MMP3*, *MMP9*, *ADAMTS4*, and *ADMATS5* genes—involved in the degradation of the extracellular matrix during OA progression [136]. Interestingly, the pharmacological activation of GPR68 by Ogerin repressed MMPs’ expression in the IL-1β-treated chondrocytes (to induce OA pathological conditions) in vitro, which suggests a protective role for GPR68 in OA [136]. In summary, GPR68’s roles in inflammation appear to be context- and cell-dependent; therefore, further dissection of its roles is warranted.

### 5.2. GPR68 in Tumor Biology

GPR68 has pleiotropic effects on tumorigenesis and has multifaceted roles in tumor cells, immune cells, and cancer-associated fibroblasts (CAFs). Originally identified from HEY cells, an ovarian cancer cell line [39], GPR68 expression is upregulated in multiple types of cancers, such as colon cancer, pancreatic cancer, and breast cancer [143,148,149]. When overexpressed, GPR68 has been shown to inhibit the migration of PC3 prostate cancer cells, HEY ovarian cancer cells, MCF7 breast cancer cells, Caco-2 colon cancer cells, and A549 lung cancer cells [150,151,152,153,154]. Cytoskeleton remodeling and the G_12/13_-Rho-Rac1 pathway were indicated in the GPR68-mediated inhibition of cancer cell migration. GPR68 overexpression in PC3 cells inhibited metastasis from the prostate to the stomach, diaphragm, and spleen of severe combined immuno-deficient (SCID) mice [150]. GPR68 overexpression has also been shown to inhibit cell proliferation and increase apoptosis in HEY and MCF7 cancer cells. One recent study demonstrated that activation of GPR68 by Ogerin, in combination with the MEK inhibitor Selumetinib, induced the death and differentiation of Schwann cells isolated from cutaneous neurofibromas through the cAMP and Ras/MAPK pathway [155]. Moreover, lenalidomide-induced GPR68 expression increased the apoptosis of myelodysplastic syndrome cells and acute leukemia cells via a calcium- and calpain-dependent pathway [156]. Altogether, the studies described above suggest that GPR68 inhibits cancer cell migration and proliferation, induces cancer cell death, and exhibits anti-tumorigenic effects.

In contrast, a separate study showed that GPR68 enhanced the expression of TRPC4 ion channels and calcium influx and promoted the migration of DAOY medulloblastoma cells [157]. Another recent study demonstrated that activation of GPR68 under acidic conditions induced cytoplasmic lipid droplet accumulation and autophagy in breast cancer cells. Depletion of GPR68 inhibited breast cancer cell growth in vitro and tumor formation in vivo [158]. It has also been shown that blocking GPR68 increases ATF4 expression and ferroptotic cell death in glioblastoma cells [159]. Thus, GPR68 can have pro- or anti-tumorigenic functions, which are dependent on the cancer cell type and context.

Besides its roles in cancer cells, studies have demonstrated that GPR68 plays an important role in tumor immune cells and fibroblasts. Subcutaneous injection of B16F10 melanoma cells into GRP68-knockout (KO) mice and wild-type (WT) mice revealed that GPR68 knockout reduced melanoma growth, suggesting that GPR68 deficiency in the host cells impedes tumor growth [160]. Another study investigated the function of host cell GPR68 in prostate cancer, wherein TRAMP-C2 and RM-9 cells were subcutaneously injected into WT and GPR68-KO mice. It was determined there was a significant reduction in tumor volume in the GPR68-KO mice in comparison to the WT mice [161]. Successively, bone marrow transplantation revealed that GPR68 expression in myeloid-derived blood cells, particularly CD11b+Gr1+ double-positive cells, was required for prostate cancer cell-induced immunosuppression to sustain tumor allografts [161]. In a B16F10 melanoma mouse model, an acidic tumor microenvironment induced the expression of GPR68 in T cells, attenuated T cell function, and promoted tumor growth. GPR68 knockout reactivated CD8+ cytotoxic T cells and enhanced the anti-tumor responses [162]. Moreover, another study showed that GPR68 inhibited the infiltration of CD8+ T cells and natural killer cells in B16F10 melanoma and promoted tumor growth in male mice but not in female mice [163]. It has also been shown that extracellular acidosis activates the proton-sensing GPR68 and GPR65 to increase the expression of the immune checkpoint PD-L1 in squamous-cell carcinoma (SCC) and B16F10 melanoma cells [119]. Together, these studies suggest that GPR68 regulates immune cell function, modulates the tumor immune microenvironment, and functions as a potential immune checkpoint to control tumor development.

Studies have revealed crucial roles of GPR68 in cancer-associated fibroblasts (CAFs) as well. Extracellular acidosis promoted the transformation of mesenchymal stem cells into CAFs through GPR68 and its downstream effector YAP [164]. In pancreatic ductal adenocarcinoma (PDAC), GPR68 expression was increased in CAFs when compared to normal pancreatic fibroblasts. The activation of GPR68 by an acidic microenvironment augments IL-6 expression via the cAMP/PKA/CREB pathway in CAFs [149]. IL-6 is a cytokine that can stimulate cancer cell proliferation and angiogenesis. Interestingly, Lorazepam, a selective GPR68 agonist, increased the expression of IL-6 through GPR68 in human PDAC CAFs and was associated with poor survival outcomes in pancreatic cancer [165]. Conophylline, a vinca alkaloid, inhibited the growth of hepatocellular carcinoma and reduced cytokine expression by CAFs through a suppression of GPR68 expression [166].

### 5.3. GPR68 Small-Molecule Modulators

As with GPR65 and GPR4, GPR68 is another one of the proton-sensing G-protein-coupled receptors for which small-molecule modulators have been discovered (Table 3). In parallel to this discovery, investigations into the development of new agonists and antagonists have been conducted in efforts to support a better understanding of the biology behind GPR68’s function in inflammation and cancer biology.

GPR68 was previously identified as a regulator of the cellular response to tissue acidosis in an infarcted heart [167]. An agonist, 3,5-disubstituted isoxazoles (Isx), was determined to promote cardiomyogenic and pro-survival gene expression via the activation of GPR68 in the subepicardial tissue around the myocardial infarction [167]. Another group screened for small-molecule modulators of GPR68, and several were discovered [122]. Notably, the benzodiazepines Lorazepam and Ogerin were determined to be potent allosteric activators of GPR68, demonstrated via several functional assays. In addition, one in vivo study in GPR68-knockout mice found that Ogerin did not impact recall in fear-conditioned mice, whereas there was a clear suppression in recall in wild-type mice [122]. Subsequent research has re-confirmed that several benzodiazepines can activate GPR68 and initiate several downstream G-protein pathways such as G_q_ and G_s_, dependent on cell type and condition [168,169]. Specifically, it was determined that Lorazepam-induced G_s_ signaling was significantly increased when GPR68-expressing HEK293 cells and primary airway smooth-muscle and fibroblast cultures were treated with media buffered to a lower pH. At a higher pH, it was determined that G_q_ signaling was dominant. Similarly, Sulazepam was found to activate G_s_ signaling in HEK293 cells as well and has been demonstrated to prevent allergen-induced airway hyperresponsiveness through GPR68 [146,169]. It was determined recently that GPR68 expression can increase B lymphocytes in peripheral blood [170]. Treatment of wild-type mice with Ogerin and Isx also separately increased B lymphocytes in peripheral blood, demonstrating that in vivo modulation of this receptor can lead to tangible changes in physiology [170]. In cancer biology, cytoplasmic lipid droplet formation has been described as an adaptation to an acidic tumor microenvironment, with this concept being demonstrated in MCF7 and T47D cells [158]. In the absence of extracellular acidosis, the activation of GPR68 in MCF7 and T47D cells using Ogerin also increased the formation of cytoplasmic lipid droplets, indicating that GPR68 may be responsible for this cellular adaptation to the acidic tumor microenvironment [158]. A separate group has confirmed that 128 (Osteocrin_33-55_), 139 (CART(42-89)_9-28_), and rat 148 (Corticotropin_17-40_) are all positive allosteric modulators of GPR68, and were all found to activate G_q_ and G_s_ signaling pathways in a cell line model. In addition, the responses to these endogenous peptides were found to cause an almost two-fold increase in activity compared to Ogerin [171]. Moreover, the same group who identified Ogerin as a positive allosteric modulator of GPR68 recently completed a structure−activity relationship study, finding that modification of the benzyl alcohol and benzylamino groups of the molecule could lead to a 33-fold increase in allosteric activity [172].

Small-molecule inhibitors of GPR68 have also been discovered, including GPR68-I and Ogremorphin. In previous reports, GPR68 was identified to promote intestinal inflammation in the IL10-/- murine model for colitis. Treatment of mice with GPR68-I relieved acute and chronic DSS-induced colitis [144]. Separately, glioblastoma multiform (GBM) is characterized by an acidic tumor microenvironment, which is hypothesized to be pro-tumorigenic. There also seems to be a positive correlation between the acidic tumor microenvironment of GBM and GPR68 expression. One group demonstrated that the inhibition of GPR68 activity using Ogremorphin promoted GBM cell death more readily than the first-line chemotherapy temozolomide (TMZ) in vitro [159]. It was also determined in another study that GPR68 can promote cardiac inflammation and fibrosis in a chronic kidney disease (CKD) model [173]. The same group found that *Cephalotaxus harringtonia* var. *nana* extract and homoharringtonine inhibited GPR68 activity and that the use of these inhibitors ameliorated CKD-induced cardiac inflammation, fibrosis, and dysfunction in a murine model [174].

Overall, there have been various new compounds identified that can modulate GPR68 activity (Table 3). In addition, recent investigations outlined here demonstrate the potential usefulness of small-molecule modulators in revealing the diverse biological functions of GPR68 in normal physiology and disease.

**Table 3 genes-15-01151-t003:** GPR68 small-molecule modulators and their biological effects.

Receptor	Compound	Mechanism	Model/Cell Type	Effects	Ref.
GPR68	Ogerin	Agonist	HEK293T cells expressing GPR68	Increases cAMP signaling	[122]
			Mouse fear-conditioning model	Suppresses recall in fear conditioning	[122]
			Primary human lung fibroblasts of healthy and idiopathic pulmonary fibrosis patients	Causes the inhibition and partial reversal of TGF-β-mediated myofibroblast differentiation	[145]
			Human cutaneous neurofibroma-derived Schwann cells	In combination with Selumetinib, induces death and differentiation of Schwann cells isolated from cutaneous neurofibromas	[155]
			C57BL/6 mice injected with Ogerin	Increases B lymphocytes in peripheral blood	[170]
			MCF7 and T47D breast cancer cells	Increases the accumulation of lipid droplets in the cells	[158]
GPR68	Sulazepam	Agonist	HEK293 cells expressing GPR68	Selectively activates cAMP signaling	[169]
			Dust-mite-induced airway hyperresponsiveness (AHR) mouse model	Reduces airway resistance	[146]
GPR68	Lorazepam	Agonist	HEK293 cells expressing GPR68	Activates cAMP signaling and calcium mobilization	[169]
			Pancreatic ductal adenocarcinoma	Is associated with decreased pancreatic cancer patient survival; increases the expression of IL-6 in cancer-associated fibroblasts	[165]
GPR68	Isx (3,5-disubstituted isoxazoles)	Agonist	Mouse myocardial infarction model with left anterior descending coronary artery ligation	Promotes cardiomyogenic and pro-survival gene expression in subepicardial tissue after myocardial infarction	[167]
			C57BL/6 mice injected with Isx	Increases B lymphocytes in peripheral blood	[170]
GPR68	128 (Osteocrin_33-55_)	Agonist	Flp-In T-REx 293 cells	Activates G_q/11_ (calcium mobilization) and G_s_ (cAMP signaling)	[171]
GPR68	139 (CART(42-89)_9-28_)	Agonist	Flp-In T-REx 293 cells	Activates G_q/11_ (calcium mobilization) and G_s_ (cAMP signaling)	[171]
GPR68	Rat 148 (Corticotropin_17-40_)	Agonist	Flp-In T-REx 293 cells	Activates G_q/11_ (calcium mobilization) and G_s_ (cAMP signaling)	[171]
GPR68	GPR68-I	Inhibitor	DSS-induced acute and chronic colitis models	Decreases disease severity, histopathologic parameters, and macrophage infiltration; reduces TNF, IL-6, and TGF-β1	[144]
			Treatment of fibroblasts and CD14+ human monocytes with extracellular acidosis	Inhibits GPR68-mediated inositol phosphate (IP) production	[129]
GPR68	Ogremorphin	Inhibitor	Glioblastoma cells	Induces glioblastoma cell death through ferroptosis	[159]
GPR68	*Cephalotaxus harringtonia* var. *nana* extract	Inhibitor	Chronic kidney disease (CKD) mouse model	Ameliorates CKD-induced cardiac inflammation, fibrosis, and dysfunction	[174]
GPR68	Homoharringtonine	Inhibitor	Chronic kidney disease (CKD) mouse model	Ameliorates CKD-induced cardiac inflammation, fibrosis, and dysfunction	[174]

## 6. Concluding Remarks

Maintaining physiological pH levels is critical for tissue homeostasis. However, many diseases can disrupt pH homeostasis, resulting in localized or systemic acidosis, which has a significant impact on cellular and/or organismal function [9,15,147,175,176,177,178,179,180,181,182,183]. Of the handful of mechanisms that support cellular responses to extracellular acidosis, the proton-sensing GPCR family has emerged as an important cell-surface receptor involved in cellular adaptation to extracellular acidosis [9,20,29,40,95]. Proton-induced activation of these proton-sensing GPCRs modulates several downstream G-protein signaling pathways, e.g., the G_s_/cAMP, G_q_/Ca^2+^/PKC, and G_13_/Rho pathways, which elicits a cellular and physiological response. There is a significant body of evidence indicating that the activation of proton-sensing GPCRs can regulate various physiological systems, including the cardiovascular, immune, renal, respiratory, skeletal, and nervous systems, which demonstrates the importance of improving our understanding of this family of receptors [14,16,27,28,50,63,108,111,131,184,185,186,187,188,189,190,191,192].

A variety of diseases, such as cancer, inflammation, ischemia, and renal, respiratory, and metabolic diseases, can result in localized tissue or systemic acidosis [1,7,15,17,18,193,194]. It is clear that proton-sensing GPCRs are involved in the pathophysiological response associated with acidosis-related diseases, particularly cancer and inflammation. Proton-sensing GPCRs have been described as tumor suppressors and promoters, which is dependent on the cancer cell type and context [63,65,111,116,150,151,161]. The functional role of proton-sensing GPCRs in inflammatory diseases is also complex. For example, proton-induced activation of GPR4 in endothelial cells triggers a pro-inflammatory signal and increases endothelial cell adhesion to leukocytes, thus supporting leukocyte infiltration into tissues and, in turn, supporting localized inflammation [14,33,34,41,44,46,59,60,61,62,71,72,74]. Alternatively, proton-induced activation of GPR65 attenuates cytokine production in leukocytes and reduces inflammation [36,93,94,97,98,105,106,127,195].

Proton-sensing GPCRs are novel targets for the treatment of various diseases, specifically those that fall under the umbrella of acidosis dysregulation, particularly because G-protein-coupled receptors are highly sought-after pharmaceutical targets and are responsible for about 30 percent of the drugs currently on the market [196,197,198,199]. Over the last decade, several synthetic small-molecule agonists and antagonists have been identified to modulate proton-sensing GPCR activity, including antagonists for GPR4 [61,62,73,74], agonists and antagonists for GPR65 [94,122,125,128], and agonists and antagonists for GPR68 [122,144,159,167,171,172,174]. These small molecules are critical pharmacological tools that can facilitate additional in vitro and in vivo investigations supporting the elucidation of the functions and signaling of proton-sensing GPCRs.

While a lot has been learned about the role of proton-sensing GPCRs in inflammation and cancer in the past two decades, there remain several areas that require exploration in future research. First, proton-sensing GPCRs have only been studied in a limited number of disease models. As pH homeostasis is associated with a wide range of pathophysiologies, it is important to elucidate the function and mechanisms of proton-sensing GPCRs in additional disease models. Second, genetic redundancy among the proton-sensing GPCR family members presents both challenges and opportunities to target these receptors. It is insightful to better understand functional similarities and differences between proton-sensing GPCR family members. Third, multiple downstream G-protein pathways are coupled to proton-sensing GPCRs. It is crucial to determine which G-protein pathways are involved in specific pathophysiological conditions. Fourth, the small-molecule modulators of proton-sensing GPCRs have previously been evaluated in preclinical models. Future clinical studies will be required to determine the efficacy and safety of the investigational compounds in relevant human diseases. Overall, proton-sensing GPCRs may have the potential to act as useful targets for the treatment of a myriad of diseases, and therefore should be investigated further.

## Figures and Tables

**Table 1 genes-15-01151-t001:** GPR4 small-molecule modulators and their biological effects.

Receptor	Compound	Mechanism	Model/Cell Type	Effects	Ref.
GPR4	**39C**, also known as EIDIP (2-Ethyl-3-(4-(E-3-(4-isopropyl-piperazin-1-yl)-propenyl)-benzyl)-5,7-dimethyl-3H-imidazo(4,5-b)pyridine)	Inhibitor	Human vascular endothelial cells	Reduces the expression of IL-1, IL-8, CXCL1, CXCL2, CCL2, CCL7, VCAM-1, ICAM-1, E-selectin, RELB, COX2, ATF3, and CHOP in cultured endothelial cells	[33,45]
			GPR4-expressing HEK and HeLa cells	Attenuates acidic-pH-induced cAMP formation	[61]
			Mouse VEGF-induced angiogenesis model	Anti-angiogenic effects	[61]
			Rat antigen-induced arthritis model	Anti-inflammatory effects, such as reduced knee swelling and joint damage/erosions	[61]
			Rat hyperalgesia model	Anti-nociceptive effects	[61]
			Short-term emphysema exacerbation COPD mouse model	Rescues manifestations of disease such as lung permeability, inflammation, mucous hypersecretion, airway remodeling, and protease secretion	[75]
			Mouse heterotopic intestinal transplant model	Reduces fibrogenesis and decreases collagen deposition	[60]
GPR4	NE-52-QQ57, also known as Compound 13 (2-(4-((2-Ethyl-5,7-dimethylpyrazolo[1,5-a]pyrimidin-3-yl)methyl)phenyl)-5-(piperidin-4-yl)-1,3,4-oxadiazole)	Inhibitor	GPR4-expressing HEK and HeLa cells	Attenuates acidic-pH-induced cAMP formation	[62]
			Cultured chondrocytes	Reduces the expression of TNF-α, IL-1β, IL-6, iNOS, nitric oxide (NO), COX2, and PGE2	[77]
			Mouse VEGF-induced angiogenesis model	Anti-angiogenic effects	[62]
			Rat antigen-induced arthritis model	Anti-inflammatory effects, such as reduced knee swelling	[62]
			Rat hyperalgesia model	Anti-nociceptive effects	[62]
			Mouse post-traumatic and aging-associated osteoarthritis models	Reduces osteoarthritis progression and promotes extracellular matrix production	[44]
			Mouse DSS-induced acute colitis model	Alleviates intestinal inflammation; downregulates cytokines such as TNF-α and adhesion molecules such as VCAM-1, E-selectin, and MAdCAM-1 in colon tissues	[46]
			Mouse IL10-deficient spontaneous colitis model	Lack of significant efficacy but exhibits a trend towards improvement for several disease parameters such as histopathology, inflammation, and hyperplasia scores	[76]
			Mouse hindlimb ischemia–reperfusion model	Reduces tissue edema, leukocyte infiltration, and exudate formation	[41]
			GPRASP1 loss-of-function mouse model of arteriovenous malformations	Reduces brain hemorrhagic lesions, cerebral edema, and arteriovenous malformations	[79]
			1-methyl-4-phenyl-1,2,3,6-tetrahydropyridine (MPTP)-induced mouse Parkinson’s disease model	Attenuates dopaminergic neuronal loss and ameliorates motor and memory functions	[80]
			Human neuroblastoma SH-SY5Y cells treated with H_2_O_2_ and MPP+	Inhibits neuronal cell death	[42]
			Mouse melanoma metastasis model	Reduces lymphangiogenesis and melanoma lymph-node metastasis	[72]
GPR4	3b (2-((2-Ethyl-5,7-dimethyl-3H-imidazo[4,5-b]pyridin-3-yl)methyl)-8-(piperidin-1-ylmethyl)-10,11-dihydro-5H-dibenzo[b,f]azepine)	Inhibitor	Mouse myocardial infarction model	Leads to prolonged survival after myocardial infarction	[73]
			Pituitary tumor cell treatment with acidic media	Reduces growth hormone and prolactin expression in MtT/S pituitary tumor cells	[81]

**Table 2 genes-15-01151-t002:** GPR65 small-molecule modulators and their biological effects.

Receptor	Compound	Mechanism	Model/Cell Type	Effects	Ref.
GPR65	BTB09089 (3-[(2,4-dichlorobenzyl)thio]-1,6-dimethyl-5,6-dihydro-1H-pyridazino[4,5-e][1,3,4]thiadiazin-5-one)	Agonist	Mouse splenocytes and peritoneal macrophages	Increases levels of cAMP; suppresses IL-2 expression of splenocytes when stimulated with anti-CD3 and anti-CD28 antibodies; suppresses TNF-α and IL-6 expression in peritoneal exuded macrophages induced by thioglycolate when stimulated with lipopolysaccharides	[94]
			Rat ischemic stroke model induced by middle cerebral artery occlusion (MCAO)	Leads to activation of the AKT pathway and protection against neuronal apoptosis and neurological deficits	[123]
			Mouse ischemic stroke model induced by photothrombotic ischemia	Inhibits microglial activation, reduces neuronal damage, and improves neurological function recovery following ischemic stroke	[124]
			Mouse hepatic macrophages	Promotes M1 macrophage polarization and stimulates the release of TNF-α, IL-6, and TGF-β	[126]
GPR65	BRD5075 BRD5080	Agonist	Mouse bone-marrow-derived dendritic cells	Reduces the expression of inflammatory cytokines and chemokines in dendritic cells	[127]
			HeLa cells expressing GPR65	Increases cAMP signaling	[127]
GPR65	ZINC13684400	Agonist	HEK293T cells expressing GPR65	Increases cAMP signaling	[122]
GPR65	ZINC62678696	Inhibitor	HEK293T cells expressing GPR65	Decreases cAMP signaling	[122]
			Mouse hepatic macrophages	Abrogates M1 macrophage polarization and reduces the release of TNF-α, IL-6, and TGF-β	[126]
			Carbon tetrachloride (CCl_4_) murine liver fibrosis model	Alleviates hepatic fibrosis and inflammation	[126]
GPR65	PTT-4256	Inhibitor	MC38 colon cancer and B16F10 melanoma mouse models	Counteracts acidic-pH-mediated immunosuppressive transcriptional programs in immune cells and shows significant therapeutic efficacy	[128]

## Data Availability

No new data were generated for this review article.

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
