# Peer review of "The Roles of Proton-Sensing G-Protein-Coupled Receptors in Inflammation and Cancer"

_genes, 2024, doi:10.3390/genes15091151_

Round 1

Reviewer 1 Report

Comments and Suggestions for Authors

The manuscript titled “The role of proton-sensing G protein-coupled receptors in 2 inflammation and cancer” by Justus et.al., is well written and provides a detailed literature review on the subject. The authors focus on GPR4, GPR65 and GPR68, elaborating their role in normal physiology, pathology, inflammation and cancer including information about their small molecule modulators.

While the information is well curated, it goes to being verbose and bordering on being hard to assimilate for the readers. Listed below are few recommendations that may help in improving the manuscript

1. It is suggested to use sub-headings to break down the huge volume of information pertaining to each GPR. The authors may follow gene description followed by

-          Role in normal physiology

-          Role in pathological conditions

This section can summarize all pathological conditions, for instance neurological, osteopathic condition and then subdivide the section to separate inflammation and cancer. Further distinctions can be made for cell autonomous and extracellular influences.

2.  Information from mouse model studies and cell lines can be a separate section to make a clear distinction between human physiology/pathology and pre-clinical investigations.

3. The mention of small molecule modulators can be solely summarized as a table rather than description in text as well as mentioned in a table.

4.  Kindly check the formatting of the table for publication purposes

5. The addition of graphical schematics with references can also reduce the verbose content to some extent and making it an easier read for the audience with more of a visual appeal.

Author Response

The manuscript titled “The role of proton-sensing G protein-coupled receptors in 2 inflammation and cancer” by Justus et.al., is well written and provides a detailed literature review on the subject. The authors focus on GPR4, GPR65 and GPR68, elaborating their role in normal physiology, pathology, inflammation and cancer including information about their small molecule modulators.

While the information is well curated, it goes to being verbose and bordering on being hard to assimilate for the readers. Listed below are few recommendations that may help in improving the manuscript

  1. It is suggested to use sub-headings to break down the huge volume of information pertaining to each GPR. The authors may follow gene description followed by

-          Role in normal physiology

-          Role in pathological conditions

This section can summarize all pathological conditions, for instance neurological, osteopathic condition and then subdivide the section to separate inflammation and cancer. Further distinctions can be made for cell autonomous and extracellular influences.

Response: Thank you for the comments.  It is a very good suggestion.  We have added sub-headings such as: (1) GPR4/65/68 in inflammatory disorders; (2) GPR4/65/68 in tumor biology; and (3) GPR4/65/68 small molecule modulators.  The sub-headings should help readers assimilate this comprehensive review focusing on the role of proton-sensing GPCRs in inflammation and cancer.

  1. Information from mouse model studies and cell lines can be a separate section to make a clear distinction between human physiology/pathology and pre-clinical investigations.

Response:  Most information about the proton-sensing GPCRs in the literature is derived from pre-clinical studies using cell culture and mouse models.  We have made a clear distinction of human patient-based studies and described so in the manuscript.

  1. The mention of small molecule modulators can be solely summarized as a table rather than description in text as well as mentioned in a table.

Response:  We believe both the text description and the table are needed and complement each other.  The text description provides a coherent flow of information, whereas the table provides distinct structured information and a quick reference for readers. 

  1. Kindly check the formatting of the table for publication purposes

Response:  We have checked the formatting of the table, which will be further checked during the typesetting process for publication.

  1. The addition of graphical schematics with references can also reduce the verbose content to some extent and making it an easier read for the audience with more of a visual appeal.

Response: With the addition of sub-headings, we believe it becomes much easier for the audience to read the manuscript.  Therefore, we feel graphical schematics may not be necessary.   

Reviewer 2 Report

Comments and Suggestions for Authors

The paper provides comprehensive review of proton-sensing G protein-coupled receptors (GPCRs), specifically GPR4, GPR65, and GPR68, and their roles in inflammation and cancer. This review concludes that proton-sensing 18 GPCRs has big therapeutic potential in pathophysiology. In general, this paper is well-written and has detailed discussion and emphasis. 

Author Response

The paper provides comprehensive review of proton-sensing G protein-coupled receptors (GPCRs), specifically GPR4, GPR65, and GPR68, and their roles in inflammation and cancer. This review concludes that proton-sensing 18 GPCRs has big therapeutic potential in pathophysiology. In general, this paper is well-written and has detailed discussion and emphasis.

Response: We thank the reviewer for the positive comments on our manuscript.

Reviewer 3 Report

Comments and Suggestions for Authors

Thank you for the opportunity to revise this interesting article. 

While overall the article is well written, sufficiently covering all aspects of the topic the authors need to make some phrasing changes prior to accepting this publication. 

-A big portion of the manuscript is the same as the published dissertation of : The Functional Roles of pH-sensing G protein-coupled receptors in Intestinal Inflammation existing at East Caroline Repository by one of the authors. Please rephrase the relevant highlighted areas by iThenticate as big portions of Chapters 1,2 and 4 are copied. Please rephrase these segments appropriately to avoid self-plagiarism. 

-A slight extension of the conclusion part and the inclusion of specific suggestions on the future areas of research rather than a general statement would be preferable. 

Comments on the Quality of English Language

Some sentences are a bit long but overall the quality of the manuscript is high 

Author Response

While overall the article is well written, sufficiently covering all aspects of the topic the authors need to make some phrasing changes prior to accepting this publication.

-A big portion of the manuscript is the same as the published dissertation of : The Functional Roles of pH-sensing G protein-coupled receptors in Intestinal Inflammation existing at East Caroline Repository by one of the authors. Please rephrase the relevant highlighted areas by iThenticate as big portions of Chapters 1,2 and 4 are copied. Please rephrase these segments appropriately to avoid self-plagiarism.

Response: Thank you for the comments.  Usually, a dissertation is not considered as prior publication.  To be sure, we contacted the editorial office about the journal policy on this issue and were told to add a note to the manuscript.  We have added a note to clarify, “Part of the text in the manuscript is from Dr. Edward J. Sanderlin's PhD dissertation” in Author Contributions.

-A slight extension of the conclusion part and the inclusion of specific suggestions on the future areas of research rather than a general statement would be preferable.

Response: We have added a paragraph to discuss the future areas of research in the “concluding remarks” section.